

# A New ZHD Model for Real-Time Retrieval of GNSS-PWV

Longjiang Li[1, 2], Suqin Wu[1, 2], Kefei Zhang[1, 2, 3], Xiaoming Wang[4], Wang Li[1, 2], Zhen Shen[1, 2], Dantong Zhu[1, 2], Qimin He[1, 2], Moufeng Wan[1, 2]

[1]Jiangsu Key Laboratory of Resources and Environmental Information Engineering, China University of Mining and Technology, Xuzhou, 221116, China
[2]School of Environment Science and Spatial Informatics, China University of Mining and Technology, Xuzhou, 221116 China
[3]Satellite Positioning for Atmosphere, Climate and Environment (SPACE) Research Centre, RMIT University, Melbourne VIC 3001, Australia
[4]Aerospace Information Research Institute, Chinese Academy of Sciences, Beijing 610209, China

*Correspondence to*: Suqin Wu (Sue.wu2018@gmail.com)

**Abstract:** The quality of the zenith hydrostatic delay (ZHD) could significantly affect the accuracy of the zenith wet delay (ZWD) of the Global Navigation Satellite System (GNSS) signal, and from the ZWD precipitable water vapor (PWV) can be obtained. The ZHD is usually obtained from a standard model – a function of surface pressure over the GNSS station. When PWV is retrieved from the GNSS stations that are not equipped with dedicated meteorological sensors for surface pressure measurements, blind models, e.g., the Global Pressure and Temperature (GPT) models, are commonly used to determine the pressures for these GNSS stations. Due to the limited accuracies of the GPT models, the ZHD obtained from the model-derived pressure value is also of low accuracy, especially in mid- and high-latitude regions. To address this issue, a new ZHD model, named as GZHD, was investigated for real-time retrieval of PWV from GNSS in this study. The ratio of the ZHD to the zenith total delay (ZTD) was first calculated using sounding data from 505 globally distributed radiosonde stations selected from the stations that had over 5,000 samples. It was found that the temporal variation in the ratio was dominated by the annual and semiannual components, and the amplitude of the annual variation was dependent upon the geographical location of the station. Based on the relationship between the ZHD and ZTD, the new model, GZHD, was developed using the back propagation artificial neural network (BP-ANN) method which took the ZTD as an input variable. The 20-year (2000−2019) radiosonde data at 558 global stations and the 9-year (2006−2014) COSMIC-1 data, which were also globally distributed, were used as the training samples of the new model. The GZHD model was evaluated using two sets of references: the integrated ZHD obtained from sounding data over 137 radiosonde stations and ERA5 reanalysis data. The performance of the new model was also compared with GPT3. Results showed the new model outperformed GPT3, especially in mid- and high-latitude regions. When radiosonde-derived ZHD was used as the reference, the accuracy, which was measured by the root mean square error (RMSE) of the samples, of the GZHD-derived ZHD, was 22% better than the GTP3-derived ones. When ERA5-derived ZHD was used as the reference, the accuracy of the GZHD-derived ZHD was 35% better than GPT3-derived ZHD. In addition, the PWV derived from 93 GNSS stations resulting from GZHD-derived ZHD was also evaluated and the result indicated that the accuracy of the PWV was improved by 23%.



## 1. Introduction

Water vapor plays an important role in both Earth's energy budget and hydrological cycle, although it only makes up 0.1 ~ 4% of the atmosphere. Therefore, accurate acquisition of water vapor is critical for both weather forecasting and climatology. During the last three decades, Global Navigation Satellite System (GNSS) has been used to retrieve precipitation water vapor (PWV), due to its high spatial-temporal resolution, all-weather, nearly real-time, high accuracy, and low cost feature. The usual procedure for obtaining GNSS-derived PWV is as following (Bevis et al., 1992): 1)

Estimating the zenith total delay (ZTD) of GNSS signals for each GNSS station; 2) Using an empirical or standard model together with surface meteorological measurements to calculate the ZHD for the GNSS station, then subtracting it from the ZTD to obtain the zenith wet delay (ZWD) of the GNSS signals for the station; 3) Converting the ZWD into PWV by multiplying the ZWD with a conversion factor which is a function of the water-vapor-weighted mean temperature ($T_m$) over the station. $T_m$ can be calculated by the approximation introduced by Askne and Nordius (1987), or from a Bevis-type model

(Bevis et al., 1992; Ross and Rosenfeld, 1997; Singh et al., 2014; Yao et al., 2014) and a blind model (Ding, 2018; He et al., 2017; Yao et al., 2012; Sun et al., 2021). The accuracies of the three types of models were analyzed in several literatures (Wang et al., 2016; Zhang et al., 2017).

      Usually, the ZHD can be determined at a millimeter-level by a standard model such as the most common model: the Saastamoinen model, under the condition that the surface pressure used in the model is measured by meteorological sensors

(Bosser et al., 2007). However, not all GNSS stations are equipped with meteorological sensors and the majority of GNSS stations are not close to any weather stations. In this case, two alternative methods are used (Wang et al., 2017): 1) using a blind model, e.g., Global Pressure and Temperature models (GPT), to obtain surface meteorological parameters for the GNSS stations; 2) using reanalysis data (e.g., ERA-Interim (Wang et al., 2017), ERA5 (Zhang et al., 2019) or NCEP (Jiang et al., 2016)) to interpolate surface meteorological parameters for the GNSS stations. Only the former can be applied for the

real-time mode.

      When real-time PWV is retrieved from the GNSS stations without meteorological sensors, the GPT model and/or its follow-up versions are usually adopted. GPT, first proposed for geodetic applications by Böhm in 2007, can provide pressure and temperature at any geographical location on the earth's surface and at any time (Böhm et al., 2007). Lagler et al. developed GPT2 by combining GPT with the Global Mapping Function (GMF), which can provide the values for more

parameters than GPT, e.g., the coefficients of the GMF (Lagler et al., 2013). In 2015, based on GPT2, Böhm et al. developed GPT2w by adding the determination of the ZWD (Böhm et al., 2015). The latest version, i.e. GPT3, was developed by Landskron in 2017, which can provide not only the parameters from GPT2w but also an empirical gradient grid (Landskron and Böhm, 2018).

      Since these models can provide pressure and temperature at any location on the earth's surface and at any time, the

blind models have been widely applied to real-time retrieval of PWV from GNSS. However, the main issue using the blind models to determine the ZHD is their limited accuracy. Wang et al. (2017) evaluated the accuracy of pressure derived from



GPT2w at 108 global GNSS stations, and found the root mean square errors (RMSE) of the pressure samples were above 7 hPa in mid- and high-latitudes regions, which resulted in large errors in PWV. A similar conclusion was made by Zhang Di (2016).

The abovementioned blind models, similar to any other empirical models, are based on the trend of the spatial-temporal variation of pressure (or the corresponding ZHD). Thus, the accuracies of the models are limited due to the dynamic feature of most atmospheric parameters. In fact, during GNSS data processing for the estimation of the ZTD and other unknown parameters, although a model-derived ZHD is sometimes used as the approximate value, it does not need to be highly accurate as the approximate value does not affect the accuracy of the final ZTD estimation results. However, when the PWV

is converted from the ZWD, which is obtained from the subtraction of the ZHD from the ZTD, the ZHD needs to be as accurate as possible for an accurate ZWD. Due to the high accuracy of the ZTD estimate, it may be used to improve the ZHD models, if the relationship between the ZHD and ZTD over the same station is known.

As is mentioned in literatures (e.g., (Luo et al., 2013; Zhang et al., 2016)), the ZHD and ZWD account for about 90% and 10% of the ZTD, respectively. However, the ratio of the ZHD to ZTD cannot be assumed about 0.9:1. In this study, the

ratio of the ZHD to ZTD was investigated using sounding data at 505 globally distributed radiosonde stations during 20-year period 2000−2019. Then, based on the relationship between the ZHD and ZTD, a new ZHD model with a good temporal resolution required by real-time retrieval of PWV was developed using the back propagation artificial neural network (BP-ANN) technique. The new model took into account not only the spatial-temporal variation in the ZHD, as current blind models do, but also the ratio of the ZHD to ZTD (i.e., the GNSS-derived ZTD was used as an input variable of the new

model).

The outline of this paper is as follows. The data used in this study are briefly introduced in section 2.1. The investigation into the ratio of the ZHD to ZTD based on sounding data at 505 globally distributed radiosonde stations is presented in section 2.2, followed by a new ZHD model developed based on the ratio of the ZHD to ZTD in the section. In section 3, the new model is validated using two sets of references: ZHDs derived from radiosonde data and ERA5 data; and

the performance of the new model is also compared with that of GPT3. The influence of the new model on PWV is also evaluated in this section. Conclusions are given in the last section.

## 2. Data and Methodologies

### 2.1 Data

Four types of data were used in this study, including sounding data from radiosonde stations, radio occultation (RO)

data from the Constellation Observing System for Meteorology, Ionosphere, and Climate (COSMIC) project, ERA5 reanalysis data from the European Centre for Medium-Range Weather Forecast (ECMWF), and GNSS data provided by the International GNSS Service (IGS). The radiosonde data were mainly used to estimate the ratio of the ZHD to ZTD and train the BP-ANN for the new ZHD model. The RO data were for improving the performance of the new model, especially over



the ocean areas. The ERA5 reanalysis data were used as a reference for the evaluation of the new ZHD model developed.
The GNSS data were used to evaluate the influence of the new ZHD model on GNSS-derived PWV. The distribution of the
four datasets and some associated information were shown in Fig. 1.

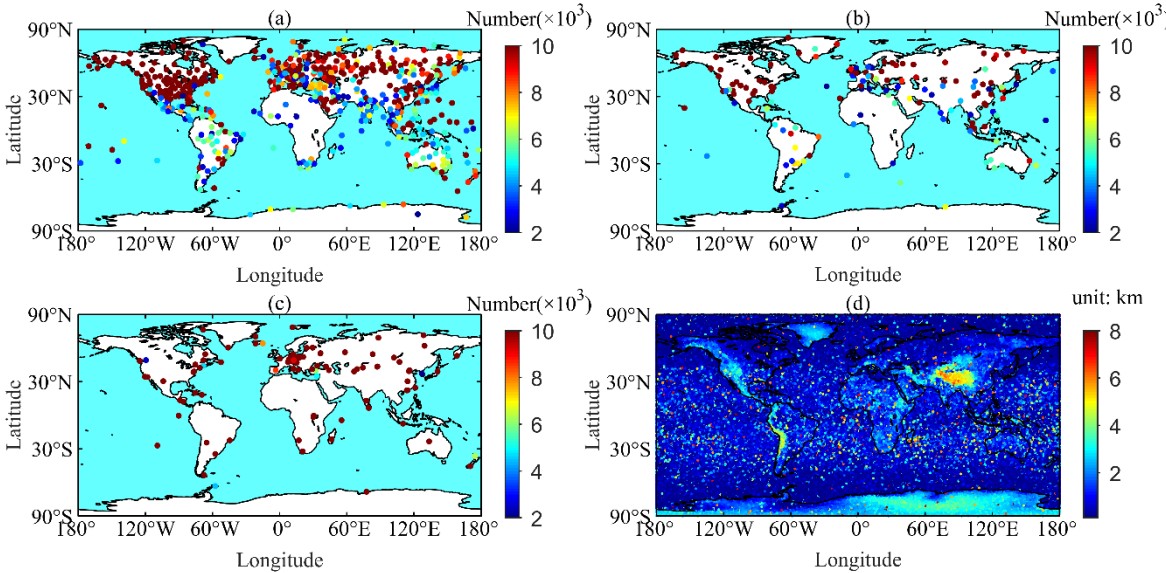

**Fig. 1 (a) Radiosonde stations for the development of the new model; (b) Radiosonde stations for testing the new model; (c) GNSS**
**stations for evaluating the effect of the new model on GNSS-derived PWV; (d) Profiles selected from COSMIC RO data. The color**
**bars in (a), (b) and (c) indicate the number of samples at each station, and the color bar in (d) denotes the penetration depth of**
**each profile.**

### 2.1.1 Radiosonde

The 20-year (2000-2019) sounding data were from the Integrated Global Radiosonde Archive (IGRA), a high-quality
radiosonde data set provided by the National Climate Data Center (NCDC). The temporal resolution of the data is usually
twice per day (or fourth times at a few stations) and the distribution of the stations included in the data set is nonuniform
(only about 1500 unevenly distributed stations are available around the world). The data set includes the observations of
pressure, temperature, geopotential height, and pressure for water vapor at the standard, surface, tropopause, and significant
levels (Durre et al., 2006). These observations form basic atmospheric profiles, based on which, the ZHD and ZTD can be
calculated by the following approximation to the definition of integral (Davis et al., 1985):

$$ZTD = \sum_{h_s}^{h_t}(k_1\frac{P_d}{T} + k_2\frac{P_w}{T} + k_3\frac{P_w}{T^2})dh \qquad (1)$$

$$ZHD = \sum_{h_s}^{h_t}(k_1\frac{P_d}{T} + k_1\frac{M_w}{M_d}\frac{P_w}{T})dh \qquad (2)$$

where $h_s$ and $h_t$ are the heights of the bottom and top layers respectively (in unit of mm, the same as the ZTD or ZHD); $P_d$
(in hPa) is the partial pressure of the dry constituent; $P_w$ (in hPa) is the partial pressure of water vapor; $T$ (in K) is the partial
temperature; $k_1$, $k_2$, and $k_3$ are the ideal gas constants from Thayer (1974).


Although a strict quality control process has been conducted on the radiosonde data from the IGRA, there still exist some missing or/and gross data in the dataset; Thus, further quality control schemes were carried out, according to the experiments and literatures (He et al., 2017; Li et al., 2020), a profile must satisfy the following seven criteria: 1) the number of pressure levels in the profile must be over 10; 2) the pressure of water vapor at the top level of the profile must be under 0.1 hPa; 3) the maximum height of the profile must be over 10 km; 4) the difference in pressures between any two adjacent

layers of the profile must be over 0 hPa and under 200 hPa; 5) the difference in heights between any two adjacent layers must be over 0 km and under 10 km; 6) the pressure levels in the profile must contain the mandatory and significant levels; 7) the number of all profiles at the radiosonde station at which the profile was obtained must be above 2,000. As a result, 695 unevenly distributed radiosonde stations over the world were identified and used in this study. Of the 695 stations, 558 stations were used as sample data, see Fig. 1(a), to develop the new ZHD model, while the other stations, shown in Fig. 1(b),

were used to test the model developed. From the two groups of stations, a total of 4,765,133 and 1,119,079 profiles were used to develop (i.e., train the BP-ANN) and test the new model, respectively. In addition, from the above 695 stations, 505 stations had more than 5,000 profiles, thus the data from the 505 stations were used to analyze the ratio of the ZHD to ZTD.

### 2.1.2 COSMIC RO

        The COSMIC initiative is one of the main RO missions, and contains COSMIC-1 and COSMIC-2 constellations. In

this study, data from the COSMIC-1, which consists of six low-earth-orbit satellites at an 800 km altitude, were adopted. During the period of 2006 to 2014, more than 4 million profiles were acquired, and the data included temperature, pressure and atmospheric density at various altitudes. Those profiles together with Eq.1 and Eq.2 were used to calculate the ZTD and ZHD.

        Traditional ZHD models such as blind models are based on harmonic functions, which need long time series data from

the same station. However, RO profiles from COSMIC are unevenly distributed over the globe, i.e., long time series data from the same site are unavailable. Hence, the traditional models are not applicable. To overcome this problem, in this study, based on the BP-ANN technique, the RO profiles were used to improve the new model, especially over the ocean regions. Considering the gross error and penetration depth (i.e., the bottom altitude) of the profiles, two criteria for the selection of valid profiles were applied. First, the penetration depth of the profile must be under 8 km, since the new model was mainly

for the earth's surface. Secondly, if the difference between the ZHDs derived from COSMIC RO data and from reanalysis data (i.e., ERA5) was above three times the standard deviation of the mean of the differences, the profile was rejected. Consequently, 3,409,654 profiles were selected to develop the new ZHD model, see Fig. 1(d) for their distribution and penetration depth (indicated by the color bar).

### 2.1.3 ERA5

ERA5 is the latest reanalysis data set provided by ECMWF and contains various hourly atmospheric variable values, including pressure, temperature, geopotential height and relative humidity at 37 pressure levels from 1000 hPa to 1 hPa, at a



specific horizontal resolution. Similar to radiosonde data, this dataset can be also used to calculate the ZTD and ZHD using Eq.1 and Eq.2.

In this study, the atmospheric data related to the ZTD and ZHD at the 37 pressure levels during the 19-year period from
2000 to 2018 over the globe were downloaded from https://climate.copernicus.eu/climate-reanalysis, and the horizontal resolution of the data is 2.5 degrees. As a relatively accurate data set, it was used as a reference to evaluate the performance of the new model developed in this study and also that of GPT3 through a comparison of the two models. In addition, the ZHDs derived from ERA5 in the 9-year period from 2006 to 2014 were also used as the reference of the ZHDs derived from the selected COSMIC RO data for the gross error identification of the RO data.

**2.1.4 GNSS**

Nowadays, over 500 global IGS tracking stations have been deployed around the world, and some of the GNSS stations are equipped with meteorological sensors, from which the relevant meteorological data can be acquired from ftp://cddis.gsfc.nasa.gov/gps/data/daily/. In addition to the meteorological data, the ZTD products provided by the IGS, as an independent data source of the training data, were also used. The ZTD and surface pressure provided by the IGS were used
to evaluate the performance of the new ZHD model dedicated to the retrieval of PWV. Since there may exist some missing and gross error observations in the meteorological data, as suggested by (Wang et al., 2007, 2017), a rigorous screening process needs to be carried out before the data are used to obtain PWV.

In this study, the surface pressure was screened using the method proposed by Wang et al. (2017). First, the pressure time series at 125 IGS stations were visually checked for deleting the stations where the pressure values had noticeably large
noises or offsets. Secondly, the pressures with the values out of the range between 550 hPa and 1,100 hPa, which were regarded unrealistic, were rejected. Thirdly, the pressure values that were larger than three standard deviations of the mean of the pressures of the station were regarded as gross errors, thus to be rejected. Finally, the stations that had less than 2,000 samples were rejected. After the above four steps were performed, 93 GNSS stations were used to evaluate the effect of the new ZHD model on the GNSS-derived PWV. The geographical locations of the 93 stations can be seen in Fig. 1(c).

**2.2 Methodologies**

In the section, the relationship between the ZHD and ZTD was first analyzed using the Lomb-Scargle periodogram at the aforementioned 505 globally distributed radiosonde stations. Then, a new ZHD model for real-time retrieval of PWV from GNSS was constructed based on the relationship and the BP-ANN technique.

**2.2.1 Ratio of the ZHD to ZTD**

As mentioned in literatures (Luo et al., 2013; Zhang et al., 2016), the ratio of the ZHD to ZTD is commonly regarded as a constant around 90%, which may not be true in some areas or cases. To investigate this, the ratios of the ZHD to ZTD





derived from sounding data at the 505 selected globally distributed radiosonde stations during the period of 20 years from 2000 to 2019 were analyzed.

Fig. 2 shows the ratio results of six radiosonde stations located in different latitude regions and their power spectral density obtained from the Lomb-Scargle periodogram. We can see significant annual periodicity with large peaks from all the six examples, and semiannual periodicity from three time series (see CAM00071082, CAM00071913 and CHM00051463) with the peaks much smaller than that of the annual periodicity. This implies that the temporal variation in the ratio is dominated by the annual periodicity. Different from the ZTD time series (Li et al., 2012b) and ZHD time series (Wang et al., 2017), the ratio time series reached the maximum in winter, and the minimum in summer, which agreed well

with the fact that PWV is higher in summer than winter, leading to larger ZWD in summer. In addition, the inter-annual variations were obvious at the three stations that are located in the equator region and southern hemisphere, possibly due to the change in the trend of PWV (climate change) in these areas.

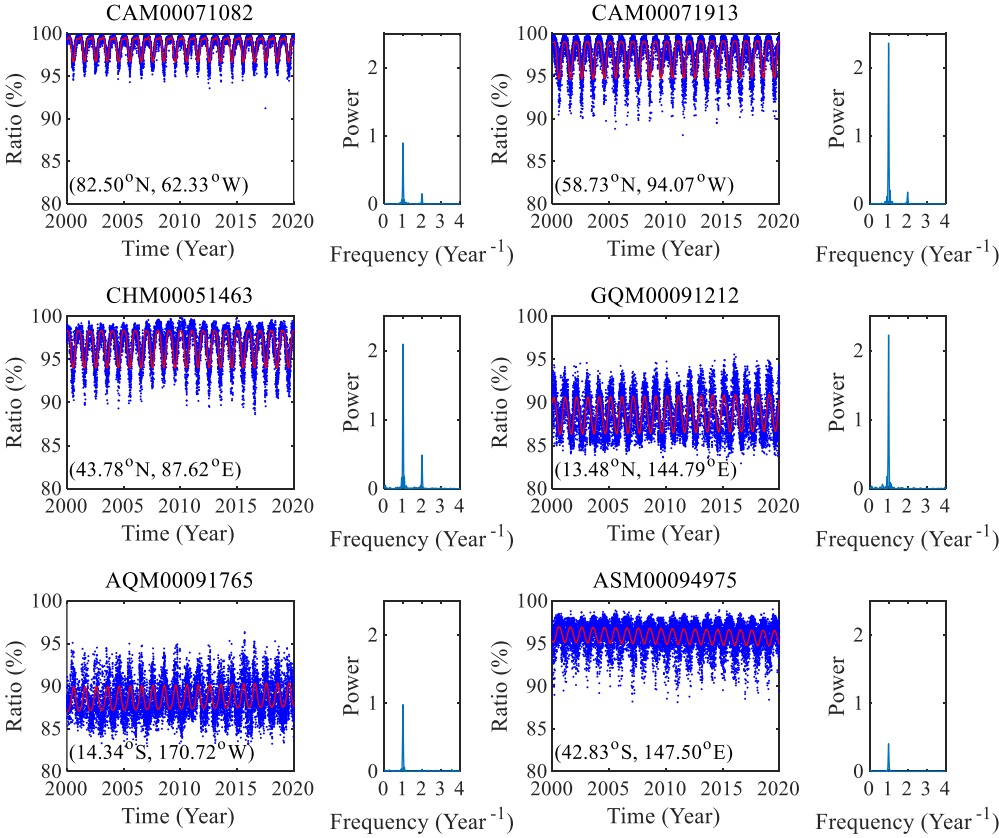

**Fig. 2 Time series of the ratio of the ZHD to ZTD at six radiosonde stations located in different latitude regions.**

In order to estimate the annual and semiannual components in the ratio time series, the following model fitting the ratio time series at each radiosonde station was adopted:





$$R = a_0 + a_1 \cdot t + a_2 \cdot \cos\left(\frac{2\pi}{365.25} \cdot t - D1\right) + a_3 \cdot \cos\left(\frac{4\pi}{365.25} \cdot t - D2\right) \quad (3)$$

where $R$ is the ratio of the ZHD to ZTD; $a_0$ is the mean of the ratio; $a_1$ is the linear trend of the ratio; $a_2$ and $a_3$ are the amplitudes of the annual and semiannual components, respectively; $D1$ and $D2$ are the phases of the annual and semiannual components, respectively; $t$ is the number of the days starting from January 1, 2000. The six unknown parameters: $a_i$ (i = 0, 1, 2, 3), $D1$ and $D2$ would be estimated using the least-squares method.

Fig. 3 shows the annual amplitude at each of the 505 stations. One can see that the annual amplitude of a station was more dependent upon the climatic type rather than latitude of the station. Most of the large annual amplitudes were found in mid-latitude regions (near 30° in both north and south hemispheres), and small annual amplitudes were found in high-latitude and the equator regions. The annual amplitudes over the eastern Atlantic and the northeast Pacific coast were small, which was likely due to the effect of the ocean (Jin et al., 2007). The annual amplitudes at all the 505 stations ranged from 0.1% to 5.7% with the mean of 2.2%. Based on the mean of the ratio of 2.2%, if the ZTD is assumed to be 2,000 mm, then the mean of the annual amplitudes in the ZHD variation is 44 mm. This is a large value, and thus to considerably affect the accuracy of GNSS-derived PWV.

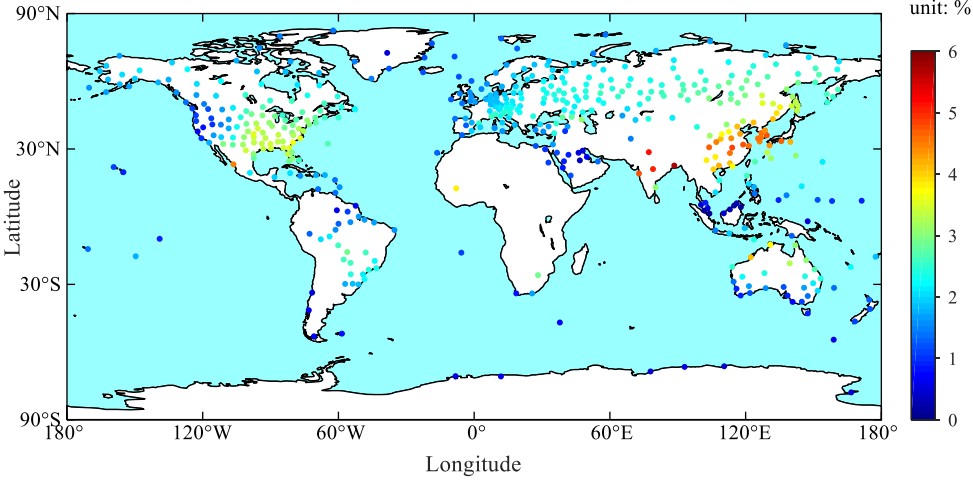

**Fig. 3 Annual amplitude of the ratio at each of 505 global radiosonde stations.**

The fact that the noticeable annual periodicity and the large annual amplitudes in the ratio time series are related to the climatic type suggests that the ratio is not always a constant, e.g., the commonly regarded 90% or any other values. Based on this characteristic, the ZHD can be obtained after the ZTD is obtained from the GNSS data processing. Thus, a new ZHD model was developed mainly for real-time retrieval of PWV from GNSS and its validation will be presented in the following sections.

### 2.2.2 The New ZHD Model

Similar to the biological neural system, the Artificial Neural Network (ANN) is a complex network composed of many neurons or nodes connecting with each other (Katsougiannopoulos and Pikridas, 2009). Its working principle is to produce



220   the target value according to the input data after being trained by the training data set. As one of the most common ANNs, the BP-ANN is a multi-layer feedbackward network trained according to the error back propagation algorithm (Li et al., 2012a). It has been applied to several fields e.g., functional approximation, pattern recognition, classification and data compression. Due to its ability of multi-parameters nonlinear regression, this study used the BP-ANN to investigate a new ZHD model, named GZHD, mainly for the real-time retrieval of PWV from GNSS.

225       The output of GZHD is the ZHD, which is required in the conversion of the GNSS-ZTD into PWV, while the input variables must be independent with each other and also related to the output. Therefore, in this study, in addition to the common five variables, i.e. the day of year (DoY), the hours of day (HoD), latitude ($\varphi$), longitude ($\lambda$) and height (H) of the station, the ZTD was also used as an input variable of GZHD based on the analysis in section 2.2.1. Using the sample data for the input variables and the output ZHD, and the BP-ANN technique, the GZHD model can be developed.

230       The BP-ANN used in this study contained three hidden layers, and the network for the new model was trained for thousands of times, depending on the number of the neurons and the activation function used in each hidden layer. From our tests, we found that the RMSE of the new model derived ZHD varied slightly (at a sub-millimeter-level) with the increase in the number of the neurons and the selection of the active function. Then, in consideration of both accuracy and efficiency of the network, the structure with 20, 20, 12 neurons and the active functions of tansig, tansig and logsig for the three hidden

235   layers were adopted. The structure of the BP-ANN including the input variables, output ZHD and three hidden layers is shown in Fig. 4.

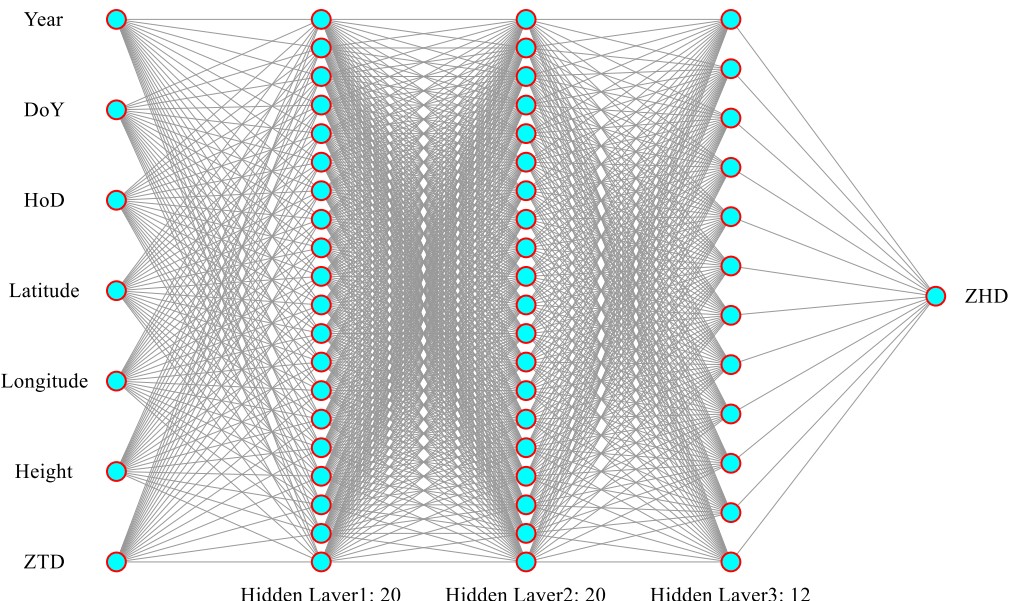

**Fig. 4 Structure of BP-ANN used in this study.**

        For the development of GZHD, two data sets were used to train the network: one was the sounding data in a period of

240   20 years at the aforementioned 558 global radiosonde stations, and the other was the COSMIC-1 RO data around the globe





in a period of 9 years. The former was taken as the main training data due to its high accuracy, while the latter was taken as auxiliary training data since the radiosonde stations were mainly deployed in continents. As mentioned before, COSMIC-1 RO profiles unevenly distribute around the globe, which are not suitable for harmonic functions used in the traditional models since long time series data from the same site are not available. However, the uneven distribution of the RO data is beneficial for the BP-ANN technique, and the usage of the RO data means an increase in the number of the training samples, which is likely to improve the performance of GZHD, especially over the ocean regions. A total of 8,174,787 RO profiles were used to train the network.

## 3 Result of GZHD

The performance of the GZHD model was assessed using the sounding data from 137 global radiosonde stations and also global ERA5 data as two reference datasets. For convenience, the ZHDs obtained from the integration expressed by Eq.2 and the data from the above two data sources were named ZHD-RS and ZHD-ERA5, respectively hereafter. The performance of GZHD was also compared with that of GPT3 by comparing the biases and RMSEs of the two model-derived ZHDs, named ZHD-GPT3 and ZHD-GZHD, respectively, based on the same reference dataset. The formulas for the bias and RMSE of the differences between the model-derived ZHDs and the references are:

$$bias = \frac{1}{n}\sum_{i=1}^{n}(ZHD_i^r - ZHD_i^m) \tag{4}$$

$$RMSE = \sqrt{\frac{1}{n}\sum_{i=1}^{n}(ZHD_i^r - ZHD_i^m)^2} \tag{5}$$

where $n$ is the number of the samples used for the evaluation; $i$ is the index of the sample; $r$ and $m$ denote the reference and model-derived, respectively. It was expected that the GZHD would outperform GPT3 since GPT3 is based on the global ZHD variation trend rather than using any actual measurements like the GZHD does (GNSS-derived ZTD is used as the input of GZHD).

### 3.1 Result using ZHD-RS as reference

ZHD-GZHD and ZHD-GPT3 calculated for each site of the above mentioned 137 global radiosonde stations during the 20-year period from 2000 to 2019 were compared against the reference of the ZHD-RS over the same station. The bias and RMSE of the ZHD-GZHD and ZHD-GPT3 over each station are shown in Fig.5.

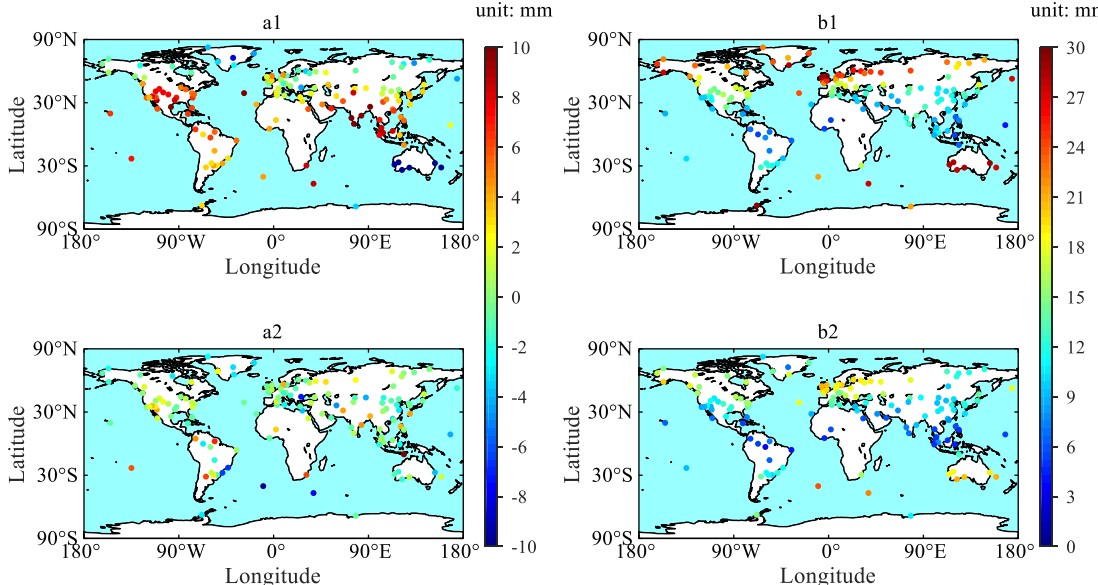


**Fig. 5 Bias (a1) and RMSE (b1) of ZHD-GPT3; and bias (a2) and RMSE (b2) of ZHD-GZHD over each of the 137 radiosonde stations during the 20-year period 2000−2019 (reference: ZHD-RS).**

In Fig. 5, subfigure (a1) indicates that the biases of ZHD-GPT3 over most stations were significant. In mid- and low-latitude regions most of the biases were positive, which was different from hight-latitude region where most biases were

negative. In addition, the biases of ZHD-GPT3 were over 10 mm (negative) at the radiosonde stations located in Australia (which will be discussed later). Subfigure (a2) shows that the bias values of ZHD-GZHD varied within ±4 mm at 119 radiosonde stations, while at the other stations (mainly distributed in the southern hemisphere) the bias values were over 4 mm. It should be noted that the number of the samples in the southern hemisphere was considerably smaller than that in northern hemisphere; thus the result of the southern hemisphere may be less trustworthy. Compared (a1) with (a2), the new

model significantly outperformed GPT3, in terms of its much smaller biases.

In subfigure (b1), the RMSE of ZHD-GPT3 appeared to be dependent upon latitude, and generally, the higher the latitude, the larger the RMSE. Most of the RMSEs in mid- and hight-latitude regions were over 20 mm, which was different from the small values in the low-latitude region. Subfigure (b2) also shows the latitude-dependent feature of the RMSE of ZHD-GZHD. However, the RMSE was smaller than that of ZHD-GPT3 at each of 123 stations, and at the other 14 stations,

which mainly distributed in the low-latitude region (a total of 45 stations), the RMSE was slightly larger than that of ZHD-GPT3. In conclusion, the accuracy of GZHD was significantly better than that of GPT3, especially in mid- and high-latitude regions.

The maximum, minimum, and mean of the biases and RMSEs of all the 137 stations shown in the four subfigures in Fig. 5 are listed in Table 1. The result indicates that the three values from the ZHD-GZHD result were much smaller than the

ones from the ZHD-GPT3 result, especially the mean values. This manifests significant improvements made by GZHD in





comparison with GPT3. In addition, the mean bias value of 2.3 mm of GPT3 implies overestimation of ZHD from the GPT3 model. In contrast, the value of −0.1 mm from GZHD means a slight underestimation of the new model.

**Table 1 Maximum, minimum, and mean of the biases and RMSEs of all the 137 stations shown in the four subfigures in Fig. 5.**

| Model | Bias (mm) | | | RMSE (mm) | | |
|-------|------|------|------|------|------|------|
|       | Mean | Min  | Max  | Mean | Min  | Max  |
| GZHD  | −0.1 | −13.4 | 11.4 | 12.3 | 3.5  | 23.7 |
| GPT3  | 2.3  | −21.6 | 13.2 | 15.7 | 4.0  | 29.8 |

For further comparison of the two model-predicted ZHD time series with the ZHD-RS time series, six radiosonde
stations were selected as examples. It is worth mentioning that the six stations are located in six continents, i.e., one radiosonde station in each of the six continents, except for Antarctica due to its very small population. The results are shown in Fig. 6, and Fig. 7 shows the correlations between the model-predicted ZHDs and ZHD-RS, named R-GZHD and R-GPT3, respectively, of the six stations.

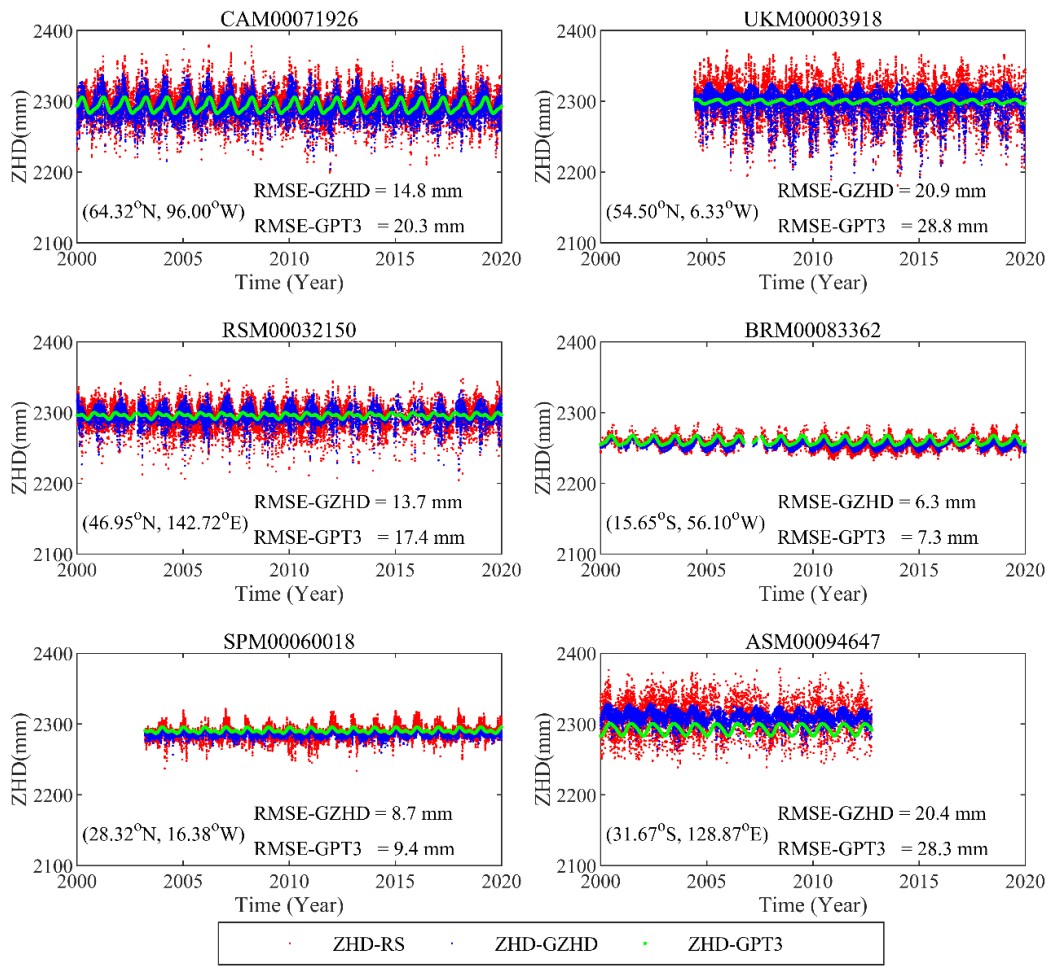

**Fig. 6 Time series of ZHD-RS (red), ZHD-GZHD (blue) and ZHD-GPT3 (green) at six radiosonde stations located in six continents.**





All the ZHD-GZHD (blue) time series in the six subfigures of Fig. 6 show not only annual and semiannual periodic variation characteristics but also high-frequency variations, which was closer to the observed (the truth) ones, compared with the ZHD-GPT3. This was because the GZHD model used the ZTD derived from sounding data as its input. In contrast, the reason for the ZHD-GPT3 time series only reflecting the annual and semiannual variations by smooth curves was that the

model was constructed based on a harmonic function that only contains two periodic terms (Landskron and Böhm, 2018). Moreover, the high-frequency variations were more significant at the stations that are located in mid- and high-latitude regions (see the first three and the last panes) than the other two, which are located in low-latitude regions. This was why GZHD significantly outperformed GPT3, especially in mid- and high-latitude regions.

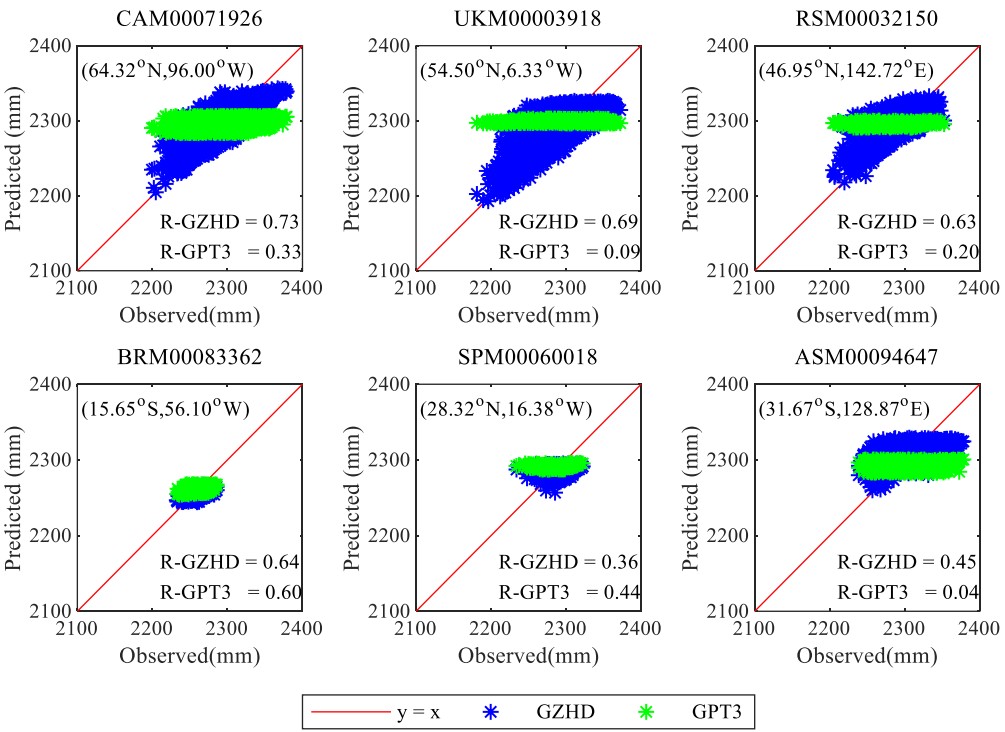

**Fig. 7 Correlation between the two model-predicted ZHD time series: ZHD-GZHD (blue) or ZHD-GPT3 (green) shown in the vertival axis, and ZHD-RS (i.e., observed, in the abscissa axis) at the six radiosonde stations shown in Fig. 6.**

From Fig. 7, we can see that in each pane the blue dots (ZHD-GZHD) distributed around the red line and much closer to the red line than the green ones (ZHD-GPT3). The result indicates that the ZHD-GZHD agreed with ZHD-RS better than ZHD-GPT3. The correlation coefficient values of R-GZHD shown in the five panes, except for the middle one in the bottom

row (which is located in Africa where only a few stations were used to construct the new model (see Fig. 1(a))), were larger than R-GPT3. This means an improvement was made by the new model. Furthermore, the improvement in a high-latitude region was more significant than a low-latitude region. For example, the R-GPT3 and R-GZHD at CAM00071926 (high-latitude) were 0.33 and 0.73, respectively, whilst the corresponding values at BRM00083362 (low-latitude) were 0.60 and 0.64, respectively.


### 3.2 Result using ZHD-ERA5 as reference

In this section, both ZHD-GZHD and ZHD-GPT3 calculated for each global grid point (with the horizontal resolution of 2.5 °× 2.5 °) during the 19-year period 2000−2018 were compared against the ZHD-ERA5 (which was not used in the construction of the new model) over the same grid. The statistical results of the 19-year data over the globe are shown in Fig. 8.

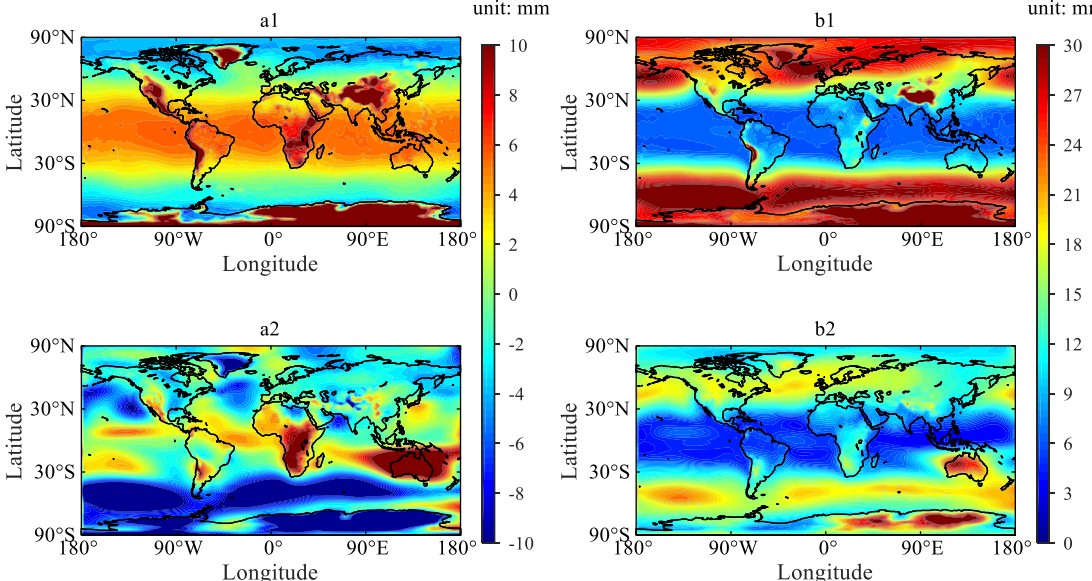

**Fig. 8 Bias (a1) and RMSE (b1) of ZHD-GPT3; and bias (a2) and RMSE (b2) of ZHD-GZHD over each global grid point (the horizontal resolution: 2.5 °×2.5 °) during the 19-year period 2000−2018 (reference: ZHD-ERA5).**

In Fig. 8, subfigure (a1) shows the latitude-dependency of the ZHD-GPT3 bias. In low-latitude regions, the biases were positive, which was different from the fact that most biases in high-latitude regions were mainly negative. The biases were also evidently positive in the Antarctica and mountainous regions such as Tibet Plateau, Andes Mountains, Rocky Mountains etc. Subfigure (a2) indicates large and positive ZHD-GZHD biases were in Africa and Australia, while large and negative biases were in Antarctica and its surrounding regions (latitude between 30°S and 90°S). Compared to the ZHD-GPT3 in (a1), the biases of ZHD-GZHD in (a2) were significantly smaller in most regions around the globe; And in Antarctica, the former were evidently negative while the latter were large and positive. This may be also due to the fact that not only the number of the samples in the southern hemisphere was considerably smaller than that in the northern hemisphere, but also the ZHD-ERA5 was of low accuracy in Antarctica (Tetzner et al., 2019; Zhang et al., 2019).

In subfigure (b1), the RMSEs of ZHD-GPT3 also appeared to be latitude-dependent, and generally, the RMSEs were higher in mid- and high-latitude regions than that in low-latitude regions. Similar to the ZHD-GPT3 biases, the RMSEs were also very large in complex mountainous terrain. The poor performance of GPT3 in complex mountainous terrain is mainly because of the mismatch between the model and actual terrain (Zhang et al., 2013; Wang et al., 2017). In subfigure (b2),



although the RMSEs were slightly dependent upon latitude, the RMSEs of ZHD-GZHD were smaller than that of ZHD-GPT3 in most regions, except for Australia and its surrounding regions (which will be discussed later). In a summary, the accuracy of the new ZHD model was higher than GPT3 in most regions when ZHD-ERA5 was used as the reference, especially in mid- and high-latitude regions.

For further evaluation of the performance of the two models in different latitudes, the mean biases and RMSEs in 12 latitude regions (with a 15° interval) were compared and corresponding results are listed in Table 2. Fig. 9 is for a better resolution (with a 2.5° interval) result. We can see that in the latitude range between 30°S and 90°N, the biases of ZHD-GZHD were smaller than that of ZHD-GPT3 in most regions. However, it was completely different in the regions in the latitude range between 30°S and 90°S, and the large bias in ZHD-GZHD was mainly because only a few radiosonde stations

were used in the construction of the GZHD model in the regions. The RMSEs of ZHD-GZHD were less than that of ZHD-GPT3 in most regions, while it was opposite in the latitude range between 30°S to 12.5°S (see Fig. 9). This was likely to be caused by the large RMSE of ZHD-GZHD in Australia (see Fig. 8). In a summary, the accuracy of ZHD-GZHD was improved by 35% in comparison with ZHD-GPT3 using ZHD-ERA5 as the reference.

**Table 2 Mean of the biases and RMSEs of ZHD-GZHD and ZHD-GPT3 during the 19-year period 2000−2018 in different latitude**
350                                        **ranges (with a 15° interval).**

| Latitude | GZHD | | GPT3 | |
|---|---|---|---|---|
| | Bias (mm) | RMSE (mm) | Bias (mm) | RMSE (mm) |
| 75°N < φ <= 90°N | −2.1 | 11.3 | −3.7 | 25.5 |
| 60°N < φ <= 75°N | −2.5 | 14.6 | −2.4 | 24.6 |
| 45°N < φ <= 60°N | −2.4 | 16.6 | −0.2 | 22.8 |
| 30°N < φ <= 45°N | −1.8 | 14.4 | 5.3 | 18.2 |
| 15°N < φ <= 30°N | −0.5 | 8.0 | 5.2 | 9.2 |
| 0° < φ <= 15°N | 2.0 | 5.4 | 5.6 | 7.1 |
| 15°S < φ <= 0° | 2.9 | 6.2 | 5.8 | 7.2 |
| 30°S < φ <= 15°S | 4.9 | 10.3 | 5.2 | 8.7 |
| 45°S < φ <= 30°S | −2.7 | 14.8 | 2.5 | 16.7 |
| 60°S < φ <= 45°S | −9.8 | 18.8 | −0.5 | 28.1 |
| 75°S < φ <= 60°S | −7.7 | 16.1 | −0.2 | 29.5 |
| 90°S <= φ <= 75°S | −9.9 | 15.8 | 15.7 | 33.2 |
| Mean | −2.6 | 12.7 | 3.4 | 19.4 |





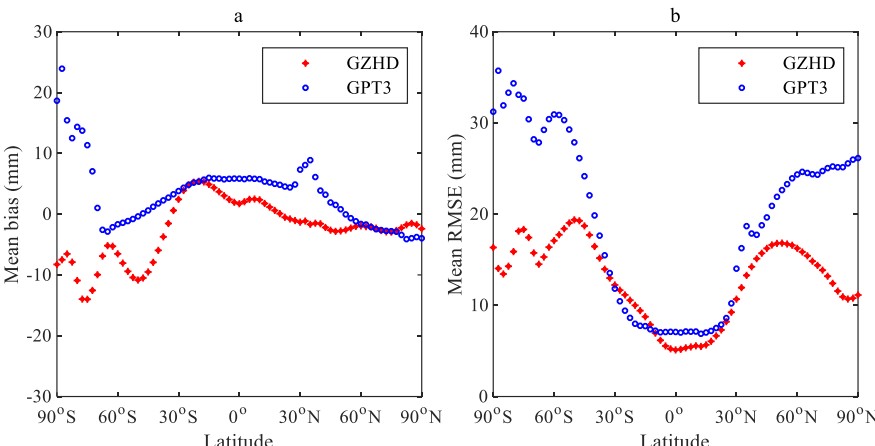

**Fig. 9 Mean biases (a) and RMSEs (b) of two model-derived ZHDs during the 19-year period 2000−2018 in different latitude ranges (with a 2.5° interval).**

It is worth mentioning that, from the results of the above two sections, in Australia, when ZHD-ERA5 was used as the
reference, the bias and RMSE of ZHD-GZHD were larger than that of ZHD-GPT3. However, when ZHD-RS was used as
the reference, the result was completely different. To investigate the cause of the difference, the two sets of ZHD reference
values at 32 radiosonde stations located in Australia were compared and results are shown in Fig. 10. It can be seen that both
the bias (in a) and RMSE (in b) were large, and over most stations the biases were negative with a (absolute) value above 10
mm, and the RMSEs over all stations were above 10 mm. The mean of all biases and RMSEs were −17.8 mm and 25.6 mm,
respectively. The large negative biases suggest a significant underestimation of ZHD-ERA5 in the region. This might be
caused by the assimilation algorithm and/or other assimilated data, although radiosonde data have been assimilated into
ERA5. When ZHD-ERA5 was used as the reference, ZHD-GPT3 agreed well with ZHD-ERA5, and much better than the
new model developed in this study, since GPT3 was based on ERA-Interim data (the last generation of reanalysis data set
provided by ECMWF).

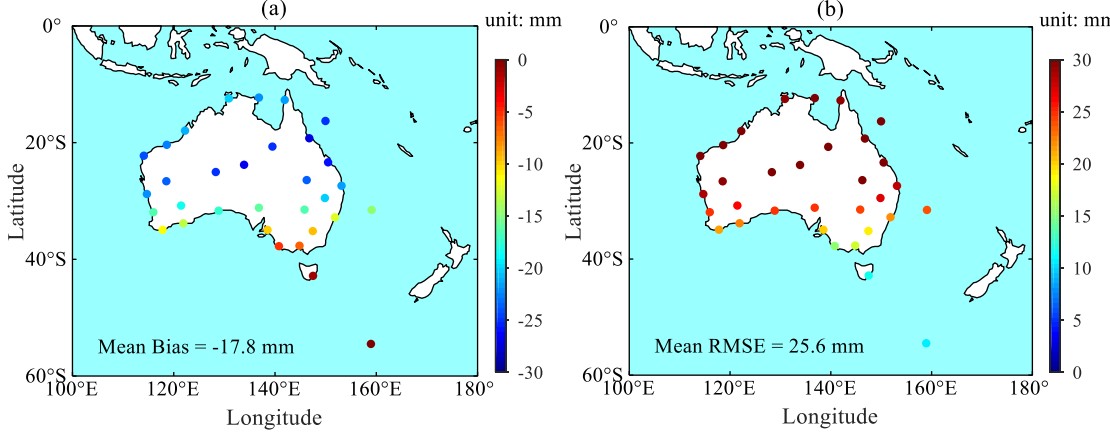


**Fig. 10 Bias (a) and RMSE (b) between ZHD-ERA5 and ZHD-RS during the 19-year period 2000−2018 over each of 32 radiosonde stations located in Australia.**





### 3.3 Result of PWV

The effect of GZHD on GNSS-derived PWV was assessed using data from 93 global GNSS stations in the 20-year
period 2000−2019. Since the accuracy of the ZHD obtained from a standard model such as the Saastamoinen model can be
at a millimeter-level under the condition that the surface pressure is measured by a meteorological sensor (Bosser et al.,
2007). In this study the PWV obtained from the high-accuracy ZHD, named PWV-OBS, was used as the reference in the
evaluation of the PWVs resulting from the ZHDs derived from the two previously tested models − GZHD and GPT3, named
PWV-GZHD and PWV-GPT3, respectively. It is noted that the ZTD provided by the IGS and $T_m$ derived from ERA5 were
used to retrieve the GNSS-derived PWVs. The bias and RMSE of the PWV-GZHD and PWV-GPT3 over each station are
shown in Fig. 11.

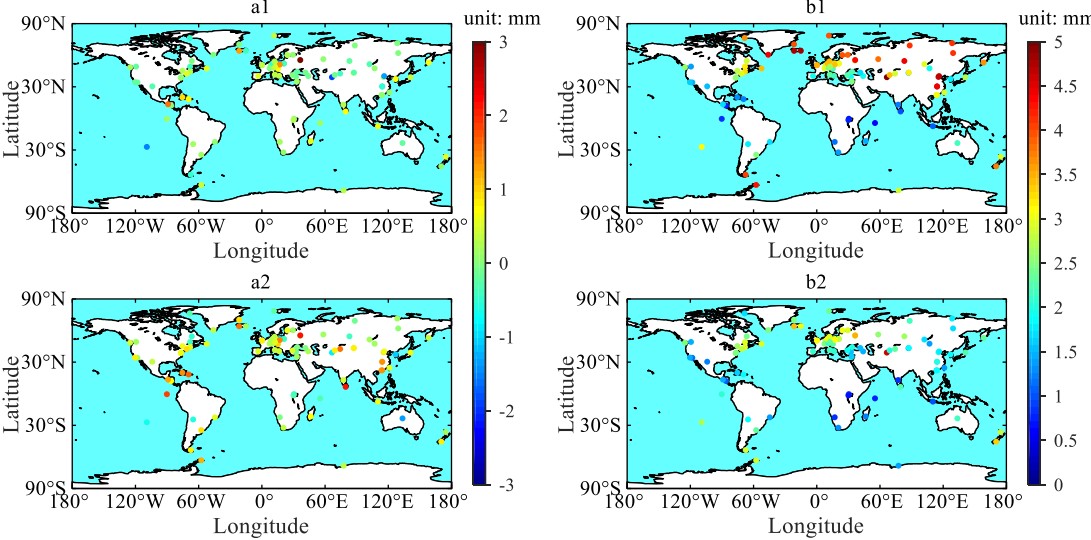

**Fig. 11 Bias (a1) and RMSE (b1) of PWV-GPT3; and bias (a2) and RMSE (b2) of PWV-GZHD in the 20-year period 2000−2019
over each of 93 global GNSS stations (reference: PWV-OBS).**

Subfigure (a1) shows the biases of PWV-GPT3 ranged from −1.9 mm to 3.0 mm with the mean of 0.1 mm; and the
biases of PWV-GZHD in subfigure (a2), ranged from −1.4 mm to 2.2 mm with the mean of 0.4 mm. Comparing the two
subfigures, one can find that at most stations the bias of PWV-GZHD was slightly larger than that of PWV-GPT3, and such
a small difference can be neglected.

Subfigure (b1) shows the RMSEs of the PWV-GPT3 varied from 0.6 mm to 5.2 mm with the mean of 2.8 mm, and the
stations where the RMSEs were large mainly distributed in mid- and high-latitude regions. Subfigure (b2) shows the RMSEs
of the PWV-GZHD varied from 0.7 mm to 4.5 mm with the mean of 2.1 mm. Although the PWV-GZHD also had a slight
dependency upon latitude, its RMSE was less than that of the PWV-GPT3 at most stations (81, of the 93 stations). The
improvement in the accuracy of the GNSS-derived PWV made by GZHD was 23%, in comparison with GPT3, which was a
significant improvement.





## Conclusion


The accuracy of the ZHD could significantly affect the quality of the ZWD from which PWV is converted using a conversion factor. The ZHD is usually obtained from a standard model – a function of the surface pressure measured by a meteorological sensor at the site of the GNSS station, and the accuracy of the ZHD is generally as high as a millimeter-level. However, not all GNSS stations are equipped with such a meteorological sensor. In addition, majority of GNSS stations are

not close to any weather stations, thus there are none surface pressure measurements available for these stations. In this case, blind models, such as a series of GPT models, are often used to obtain surface pressures. As a result, the accuracy of the model-derived ZHD is limited, especially in mid- and high-latitude regions. To address this issue, a new ZHD model was developed in this study using the following technique.

First, the ratio of the ZHD to ZTD was analyzed using the Lomb-Scargle periodogram at 505 global radiosonde stations

at each of which the number of samples was over 5,000. Their ratio time series showed significant annual and semiannual periodicities, and the annual amplitude was related to the geolocation of the station. Then, a new ZHD model, GZHD, was developed using the BP-ANN technique and sounding data from 558 global radiosonde stations together with RO data from COSMIC-1. In the GZHD model, not only the seasonal and spatial variation in the ZHD, but also the relationship between the ZHD and ZTD, were taken into consideration. More specifically, the ZTD was used as an input variable for the network

for the modelling.

The newly developed GZHD model was assessed using two sets of references: ZHD-RS and ZHD-ERA5, and the performance of the model was also compared with GPT3. Results showed that the new model significantly outperformed GPT3, especially in mid- and high-latitude regions; and the improvements in the accuracy of the ZHD-GZHD were 22% and 35% in comparison with ZHD-GPT3 based on the references of ZHD-RS and ZHD-ERA5, respectively. In addition, the

effect of the ZHD-GZHD on PWV retrieved from 93 global GNSS stations that are equipped with meteorological sensors was also evaluated using PWV-OBS as the reference. Results showed that, compared with PWV-GPT3, the accuracy of the PWV-GZHD was improved by 23%, which is significant. These results suggest the promising potential of the GZHD model for a better GNSS-derived PWV for the GNSS stations that are not equipped with meteorological sensors, especially for the real-time mode.

Our future work will be using ERA5 data in the construction of the new model to improve the performance of the new model in the southern hemisphere.

**Data availability:**

Radiosonde data: http://weather.uwyo.edu/upperair/

COSMIC RO: https://cdaac-www.cosmic.ucar.edu/cdaac/index.html

ERA5 reanalysis: https://climate.copernicus.eu/climate-reanalysis

GNSS: https://cddis.nasa.gov/Data_and_Derived_Products/GNSS/GNSS_data_and_product_archive.html





**Author contributions**

Longjiang Li designed the experiments and wrote the original draft. Suqin Wu, Kefei Zhang, Xiaoming Wang and Wang Li reviewed and edited the manuscript. Zhen Shen, Dantong Zhu, Qimin He and Moufeng Wan processed the RO data, radiosonde data, GNSS data and the ERA5 reanalysis data, respectively.

**Competing interests:**

The authors declare that they have no conflict of interest.

**Acknowledgements:**

This work was funded by the National Natural Science Foundation of China (41730109, 41874040), the Natural Science Foundation of Jiangsu Province (Grant No. BK20200646) and the Fundamental Research Funds for the Central Universities (Grant No. 2020QN31). The authors would also like to acknowledge the support of the Xuzhou Key Project (Grant No. KC19111) awarded in 2019 and the Jiangsu dual creative talents and Jiangsu dual creative teams program projects of Jiangsu Province, China, awarded in 2017. We would like to thank ECMWF, NCDC, UCAR and IGS for providing ERA5 reanalysis data, radiosonde data, RO data and GNSS data, respectively.

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
