# Peer review of "A New ZHD Model for Real-Time Retrieval of GNSS-PWV"

_Atmospheric Measurement Techniques, 2021_

## Author Comment (AC1)

**Reviewer's comments:**

For the derivation of precipitable water vapour (PWV) from GNSS zenith total delays, it is important to have accurate values of zenith hydrostatic delays (ZHD). In the best case, the ZHD are calculated from measured pressure values at the sites. In case those measurements are not available and the PWV need to be determined in near real-time, users often refer to empirical models, such as those from the GPT series. However, these blind models are not able to capture real pressure variations; thus, significant errors can show up in PWV. Consequently, the authors have developed a GZHD model based on the back propagation artificial neural network where they use measured zenith total delays to get improved values for the ZHD. With comparisons against radiosonde data and ERA5 they show that the new GZHD models provides improved ZHD values with respect to GPT3.

While I like their approach and find it very interesting, the motivation is not that clear. I would assume that forecast numerical weather models with pressure values will be available to users investigating the near real-time determination of PWV, so line 54 is not entirely correct. These forecast pressure values will be more accurate than the values from the GZHD model.

But as I said, I find the approach itself very interesting using the ratio between total and hydrostatic zenith delays as key parameter and an artificial neural network. I have not seen that before.

The biases in Figure 8 for GPT3 are rather large and systematic. Which 2.5 x 2.5 degree topography did the authors use and how did they interpolate within the ERA5 profiles?

Figure 3 caption: Ratio of total and hydrostatic zenith delays.

110 four times

277 high-latitude

305 vertical axis

**My response:**

Thanks for the reviewer's suggestions and we have made changes for all the suggestions in the new manuscript accordingly. The detailed modifications are as follows.

1) The motivation of the study includes:

    a) Improving the blind models used in real-time retrieval of GNSS-PWV under the condition that meteorological sensor are not equipped at the GNSS stations.

    b) Providing an alternative method in consideration of the inconvenience in acquiring forecast pressure from numerical weather models. The forecast data need to be downloaded in advance, which increases the complexity of data processing, not to mention the fact that the forecast data may not be obtained due to various reasons, e.g. problems of some servers or agencies.

    c) Coming up with a new method for the determination of the ZHD in the retrieval of GNSS-PWV.

2) Line 54 has been corrected.

3) There are no topography and interpolation used in Figure 8. When ZHD-ERA5 was used as the reference, the ZHD-ERA5 at a grid point was calculated by the integral from the lowest level (1000 hPa) to the highest level (1 hPa), and the ZHD-GPT3 at the same grid point was calculated using the Saastamonien model with the input pressure derived from GPT3 at the lowest level. Note that the geopotential height in ERA5 was converted to ellipsoidal height using the method described in the following paper:

*Wang, X., Zhang, K., Wu, S., Fan, S., and Cheng, Y.: Water vapor-weighted mean temperature and its impact on the determination of precipitable water vapor and its linear trend: Water Vapor-Weighted Mean Temperature, J. Geophys. Res. Atmos., 121, 833–852, https://doi.org/10.1002/2015JD024181, 2016.*

The reason for the large biases in Figure 8 may be that the GPT3 model performed poorly in the regions where the height is significantly different from the earth surface. The height of the grid points from ERA5 was above the earth surface in most regions but below the earth surface in mountainous regions, since the pressure at the lowest level of the grid points from ERA5 is almost constant (1000 hPa) but the heights at the lowest level in different grid points are different.

4) Figure 3 caption has been corrected, as advised.

5) Lines 110, 277 and 305 have been corrected, as advised.

6) In addition to the above suggestions, we also found a new problem: when the integral method was used to calculate the ZHD, geoidal height or ellipsoidal height should be used, so the geopotential height contained in ERA5 and sounding data at each pressure level needs to be converted to geoidal or ellipsoidal height. However, in the previous manuscript, the height was not converted. Although this fault might cause large biases in the ZHD, it is not necessarily to largely affect the RMSE of the ZHD. This is due to that the effect of this fault changes biases from positive to negative, but the values are about equal. In the new manuscript, all the problems have been corrected.

---

## Author Comment (AC2)

**Reviewer's comments:**

The authors established a GZHD model using BP-ANN method trained by globally distributed radiosonde data and COSMIC-1 data. Then the GZHD model was evaluated. However, the evaluations have a problem: the period of test data repeated the period of training data. Therefore the performance of GZHD model at the time away from the training period is unknown. No information about the real-time availability of GZHD model is shown. According to my experience in training this type of data-driven model, the evaluations in the study may overestimated the accuracy of the new model. Therefore detailed evaluations in the period different from the training dataset, which should be more important than the evaluations presented by the authors, must be added.

Line 52: Add a reference of GPT.

Line 100: The color in figure 1(d) agrees very well with the terrain, e.g. the Tibet Plateau. Please explain it.

**My response:**

Thanks for the reviewer's suggestions and the modifications made accordingly in the new manuscript are as follows.

1) The GZHD model was evaluated using out-of-sample data: the radiosonde and ERA5 data during the one-year period of 2020 (in which no data were used to train the model). Results showed that the performance of GZHD was almost the same as that in the training period (please see Section 3.3 in the new manuscript).

2) The effect of the GZHD model was tested on real-time GNSS-PWV and results showed that GZHD outperformed GPT3 (please see Section 3.4 in the new manuscript). Note that the real-time ZTD used to retrieve GNSS-PWV was for 154 days in 2020, and it was processed by a modified BNC software package, more details can be found in the following paper:

*Sun, P., Zhang, K., Wu, S., Wang, R., and Wan, M.: An investigation of real-time GPS/GLONASS single-frequency precise point positioning and its atmospheric mitigation strategies, Meas. Sci. Technol., https://doi.org/10.1088/1361-6501/ac0a0e, 2021b.*

3) A reference of GPT has been added in line 52.

4) The color in Figure 1(d) agrees very well with the terrain, e.g. the Tibet Plateau. This is because the penetration depth of a profile has to be above the earth surface. In addition, there are over 3,000,000 profiles in this figure, thus some of the profiles are covered by others.

5) In addition to the above suggestions, we also found a new problem: when the integral method was used to calculate the ZHD, geoidal height or ellipsoidal height should be

used, so the geopotential height contained in what data?? at each pressure?? level needs to be converted to geoidal or ellipsoidal height. However, in the previous manuscript, the height was not converted. Although this fault might cause large biases in the ZHD, it is not necessarily to largely affect the RMSE of the ZHD. This is due to that the effect of this fault changes biases from positive to negative, but the values are about equal. In the new manuscript, all the problems have been corrected.